# Different Prognostic Values of Tumour and Nodal Response to Neoadjuvant Chemotherapy Depending on Subtypes of Inflammatory Breast Cancer, a 317 Patient-Study

**DOI:** 10.3390/cancers14163928

**Published:** 2022-08-15

**Authors:** Maximilien Rogé, Julia Salleron, Youlia Kirova, Marin Guigo, Axel Cailleteau, Christelle Levy, Marianne Leheurteur, Rafik Nebbache, Eleonor Rivin Del Campo, Ioana Lazarescu, Stéphanie Servagi, Maud Aumont, Juliette Thariat, Sébastien Thureau

**Affiliations:** 1Department of Radiation Oncology, Henri Becquerel Cancer Center, 76000 Rouen, France; 2Department of Biostatistics, Institut de Cancérologie de Lorraine, 54500 Vandœuvre-Lès-Nancy, France; 3Department of Radiation Oncology, Institut Curie, 75005 Paris, France; 4Department of Radiation Oncology, Centre François Baclesse, 14000 Caen, France; 5Department of Radiation Oncology, Institut de Cancérologie de l’Ouest, Nantes, 44300 Saint-Herblain, France; 6Department of Medical Oncology, Centre François Baclesse, 14000 Caen, France; 7Department of Medical Oncology, Centre Henri Becquerel, 76000 Rouen, France; 8Department of Radiation Oncology, Tenon University Hospital, Sorbonne University, 75020 Paris, France; 9Department of Radiation Oncology, Centre de la Baie, 50300 Avranches, France; 10Department of Radiation Oncology, Institut Jean Godinot, 51100 Reims, France; 11Department of Radiation Oncology and Nuclear Medicine, Henri Becquerel Cancer Center, and QuantIF LITIS, 76000 Rouen, France

**Keywords:** inflammatory breast cancer, triple-negative, neoadjuvant chemotherapy, radiotherapy, pathological response, survival

## Abstract

**Simple Summary:**

Inflammatory breast cancer is a rare entity associated with a poor prognosis, especially for the triple-negative subtype. This study investigates the independent prognostic value of tumour and nodal responses after neoadjuvant chemotherapy. It shows that tumour and lymph node responses did not have the same prognostic value regarding HR and HER2 statuses. We identified a subgroup of patients with triple-negative inflammatory breast cancer with residual lymph node disease for whom adjuvant treatment intensification may be worth investigating.

**Abstract:**

Inflammatory breast cancer (IBC) is a rare entity with a poor prognosis. We analysed the survival outcomes of patients with nonmetastatic IBC and the prognostic value of tumour or nodal responses to assess their individual prognostic impact across IBC subtypes. This retrospective multicentre study included patients diagnosed with IBC between 2010 and 2017 to account for advances in neoadjuvant systemic therapies and modern radiotherapy at seven oncology centres in France. Three hundred and seventeen patients were included and analysed. After a median follow-up of 52 months, the 5-year DFS was lower for triple-negative (TN) (50.1% vs. 63.6%; *p* < 0.0001). After multivariate analyses, incomplete nodal response was the only significant prognostic factor in the triple-negative group (HR:6.06). The poor prognosis of TN-IBC was reversed in the case of nodal response after neoadjuvant chemotherapy. Breast response does not appear to be a decisive prognostic factor in patients with TN-IBC compared to lymph node response. Despite improvements in neoadjuvant treatments, IBC remains associated with a poor prognosis. In TN-IBC patients, lack of pathological complete node response was associated with poorer survival than any other group. Treatment intensification strategies are worth investigating.

## 1. Introduction

Inflammatory breast cancer (IBC) is a rare breast cancer subtype that accounts for 2% of invasive breast cancers [1]. The 5-year overall survival of patients with localised IBC has been improved with the introduction of neoadjuvant chemotherapy [2,3], including anthracyclines [4,5], taxanes [6,7], and HER2-directed monoclonal antibodies [8]. However, the 5-year overall survival rates remain poor, ranging from 51.6% to 69.4% [9,10]. The pathological complete response (pCR) has been reported to be a predictor of disease-free survival or overall survival. Therefore, pCR has been adopted as the primary endpoint for neoadjuvant trials [8,11]. It has been shown that the surrogate of pCR for survival may be better in aggressive breast cancer subtypes [12]. A recent meta-analysis of clinical trials investigating neoadjuvant chemotherapy in breast cancer by Cortazar et al. [13] suggests that pCR may be a better surrogate of survival when accounting for both primary pCR and nodal pCR, rather than the sole use of primary pCR.

Moreover, this concept suggests the existence of heterogeneities in the assessment of responses, with the knowledge that several classifications exist and are variably used across institutions. It also indicates that pCR is particularly relevant for patients with triple-negative breast cancer. However, the data are not specific to IBC, and the prognostic value and surrogacy of partial pathological response have rarely been addressed. However, in aggressive breast cancer (and in IBC in particular), an incomplete response is likely, and it may determine the need for maintenance or salvage therapy.

Due to its rareness, the current literature on nonmetastatic IBC mainly consists of small, monocentric, retrospective studies, as well as older case series or large (but heterogeneous) case series based on registries [9,10,14,15]. Several studies have already evaluated the impact of pCR on survival outcomes across different IBC subtypes (HR and HER2) [15,16]. However, to our knowledge, no studies have independently investigated the prognostic value of tumour or nodal responses after neoadjuvant chemotherapy according to IBC subtype. Therefore, in this large, multicentre, retrospective study conducted over a relatively short time span (which allowed the integration of up-to-date systemic and local therapies), we aimed to analyse the survival outcomes of patients with nonmetastatic IBC. We also independently investigated the prognostic value of tumour or nodal responses for triple-negative IBC in order to identify different prognostic groups in consecutive unselected patients.

## 2. Methods

### 2.1. Study Design and Participants

We used institutional registry databases to identify and extract data from women (aged > 18 years) who had histologically confirmed breast cancer of inflammatory presentation, as defined according to the 8th edition of UICC-AJCC (Union for International Cancer Control-American Joint Commission on Cancer) TNM, diagnosed between January 2010 and December 2017 at seven private and public oncology centres in France [17]. Diagnosis of inflammatory breast cancer was usually based on clinical findings (rapid onset of symptoms, erythema, “peau d’orange”, breast oedema or swelling …). Breast cancer was always confirmed with a biopsy. Pathological confirmation of dermal lymphatic emboli (skin punch biopsies) was usually not performed to confirm the diagnosis. Patients who had distant metastases at the time of diagnosis, with a follow-up period of fewer than 6 months (except for death occurring 6 months after diagnosis), who lacked the clinical findings of IBC, for whom the pathological response was not available because they did not undergo surgery or for whom HR/HER2 statuses were not available were excluded from this study. Institutional review board approval was obtained from each hospital centre participating in the multicentre retrospective data reviews (Institutional Review Board number: 2007B). The consent of each participant was handled according to the data protection officers’ requirements at each participating centre. Data were collected via a GDPR (General Data Protection Regulation) compliant encrypted secured electronic case report form (https://www.easy-crf.com, accessed on 22 April 2020).

### 2.2. Procedures

All of the patients were classified into four subgroups, according to the results of immunohistochemistry (hormonal receptor [HR] and human epidermoid growth factor receptor-2 [HER2]) that was assessed in the primary tumour from a biopsy before any therapy, as follows: HR+/HER2-, HR+/HER2+, HR-/HER2+, or HR-/HER2- (triple-negative). Local pathologists defined the hormone receptor status with immunohistochemistry (deemed positive if oestrogen receptor, progesterone receptor, or both were ≥10%).

Considering that we aim to focus specifically on the triple-negative subtype, known for its high aggressivity and limited therapeutic arsenal, we chose to group HR+/HER2+, HR-/HER2+ and HR+/HER2 subtypes as HR+ and/or HER2+ tumours to facilitate the statistical analysis and to improve the understanding of this study.

We recorded the responses after neoadjuvant chemotherapy with two pathological response assessment systems (absolute and relative grading systems), as reported by pathologists at each participating centre (ypTNM [17] and Sataloff [18]). Sataloff’s classification was the standard evaluation used by pathologists in France during the study period. Details of this classification are available in the Appendix A.

We performed our statistical analyses by using Sataloff’s classification to independently assess the impact of pathological tumour and lymph node responses on survival in concordance with other studies on inflammatory breast cancer [19].

The definition of pCR has evolved across trials and grading systems. In our study, we reported patients as having a pCR when surgical samples showed a response defined as Sataloff TA and NA or NB, according to published data [19,20]. We also used the ypTNM system and designated patients as having a pCR when surgical samples showed responses defined as ypT0/Tis and ypN0, according to the study by Cortazar et al. [13].

Regular follow-up visits were performed every six months from the end of radiotherapy until at least five years afterwards. A clinical examination and annual contralateral mammography +/− ultrasound were performed. Asymptomatic patients did not systematically undergo whole-body imaging tests.

Patient charts were reviewed to determine the dates of the last follow-ups and to document death and locoregional or distant recurrences.

The International Expert Panel published recommendations stating that all patients should receive upfront chemotherapy or chemotherapy and targeted therapy [2,3]; therefore, we focused our main analysis on patients who received neoadjuvant chemotherapy in our population of consecutive patients who were treated in routine practice.

### 2.3. Statistical Analysis

Quantitative parameters were described by the means and standard deviations or by the medians and interquartile ranges [IQRs], according to the normality of the distribution as assessed by the Shapiro–Wilk test. Qualitative parameters are expressed as frequencies and percentages. Demographics and clinical characteristics were compared between HR+ and/or HER2+ and TN groups by using a chi-square test or Fisher’s exact test for qualitative parameters and a Mann–Whitney U test or Student’s *t*-test for quantitative parameters. Disease-free survival (DFS), overall survival (OS), distant metastases-free survival (DMFS), and locoregional relapse-free survival (LRFS) were described with the Kaplan–Meier method and compared between the two groups via a univariate Cox proportional hazards model. Survival analyses were censored at 5 years in accordance with the median duration of follow-up. The interaction term between Sataloff T and the two groups was tested to determine whether Sataloff T had the same prognostic impact on DFS across the groups. The same analysis was performed for Sataloff N. The impact of patient characteristics on DFS was subsequently stratified by TN status.

For each group (HR+ and/or HER2+ and TN groups), the following analysis was performed: first, each characteristic was tested by using a univariate Cox proportional hazards model. Parameters with a *p*-value less than 0.1 were included in a multivariate Cox proportional hazards model. To investigate the stability of the final model, internal validation with the bootstrap resampling method was used. The results were described using the hazard ratio (HR) and its 95% confidence intervals (CI). The model assumption was checked by using statistical tests and graphical diagnostics based on the scaled Schoenfeld residuals. We also performed the same previous analysis on overall survival.

As an exploratory analysis, we performed a recursive partitioning analysis (RPA) on the significant parameters found in the previous multivariate analyses to define different prognostic groups for DFS. The most significant variable in the multivariate model was selected as the parent node to generate a survival tree for DFS. This parent node would split into child nodes as homogenous as possible to dependent variables. The RPA evaluated all of the possible dichotomous splits for all of the potential prognostic factors and then chose the split providing the most separation between the two groups (children nodes) with respect to DFS. The goodness-of-split criteria were evaluated via the log–rank statistic. We also used a nonparametric bootstrap with resampling to perform internal validation of our findings.

All of the statistical analyses were performed by using SAS software v9.4 (Institute Inc., Cary, NC, USA). *p*-values < 0.05 were considered to be statistically significant.

### 2.4. Outcomes

The primary endpoint was the 5-year DFS, which was defined as the time from diagnosis to recurrence (locoregional and/or distant metastases) or death (all causes). Secondary endpoints evaluated the prognostic value of the primary tumour or nodal responses after neoadjuvant chemotherapy to identify different prognostic groups among patients without a pCR. Other secondary endpoints were the 5-year overall survival (OS), 5-year locoregional relapse-free survival (LRFS), 5-year distant metastases-free survival (DMFS), and proportion of patients with pCR as defined in procedures.

## 3. Results

### 3.1. Population

Of the 364 patients who were diagnosed with nonmetastatic IBC and screened for inclusion in our study, those patients receiving neoadjuvant chemotherapy (*n* = 346, 95%), whose HR and HER2 statuses were available (*n* = 337, 93%) and who underwent surgery (*n* = 317, 87%) were included (Appendix A). Details of the excluded patients (*n* = 47, 13%) are available in the Appendix A.

The baseline demographics and clinical characteristics of the 317 analysed patients are shown in Table 1. The median age was 53 years old. One-third of the patients had a body mass index (BMI) ≥ 30 (*n* = 108, 34.2%). Most of the patients had ductal carcinoma (88.9%) and nodal metastases at diagnosis (83.8%). Ninety-nine patients (31.2%) had HR- and HER2- tumours, i.e., triple-negative (TN) tumours, and 218 patients (68.8%) had HR+ and/or HER2+ tumours. Patients with TN tumours were different from patients having HR+ and/or HER2+ tumours. Specifically, the patients had incidences of Scarf Bloom Richardson (SBR) grade 3 (77.1% vs. 46.5%; *p* < 0.001), Ki67 > 30% (86.0% vs. 52.1%; *p* = 0.0002), and clinical lymph nodes (cN+) (89.9% vs. 81.0%; *p* = 0.047) more frequently than patients with HR+ and/or HER2+ tumours. Clinical characteristics for patients with HR+ and/or HER2+ tumours according to the three subtypes (HR+/HER2+, HR+/HER2-, HR-/HER2+) are available in the Appendix A.

According to our criteria defined in the methods section, all patients had received neoadjuvant chemotherapy (Table 2). Most patients receiving neoadjuvant chemotherapy underwent a mastectomy and axillary lymph node dissection (*n* = 310, 97.8%), followed by adjuvant radiotherapy (*n* = 291, 91.8%). Neoadjuvant chemotherapy included anthracyclines in 284 patients (89.6%) and taxanes in 316 patients (99.7%). Eighteen patients (5.7%) received preoperative rescue radiotherapy after progression under neoadjuvant chemotherapy. Seven patients with TN tumours received adjuvant capecitabine. The sequence of the different strategies that were used is summarised in the Appendix A.

### 3.2. Responses to Treatment

The evaluation of responses after neoadjuvant chemotherapy using Sataloff and ypTNM is shown in Table 1. There was no significant difference in responses using the Sataloff T or Sataloff N systems between HR+ and/or HER2+ tumours compared to TN tumours. According to the Sataloff definition of pCR, 95 patients (30.4%) achieved a pCR with no significant difference between patients having HR+ and/or HER2+ tumours compared to TN tumours (60 [27.8%] versus 35 [36.5%]; *p* = 0.12). According to the ypTNM system after neoadjuvant chemotherapy, 84 (26.5%) patients achieved pCR, 51 (23.4%) patients had HR+ and/or HER2+ tumours, and 33 (33.3%) patients had TN tumours (*p* = 0.063). As expected and shown in Appendix A, there are significant disparities in response (Sataloff and ypTNM) among the three subtypes of HR+ and/or HER2+ tumours.

### 3.3. Follow-Up

The median follow-up was 52.5 months (interquartile range: 35–77) and 64 months in the surviving patients (interquartile range: 42–81). The 5-year DFS was 59.0% (95% CI: [53.0; 64.6]) and was significantly lower for TN tumours (50.1%; 95% CI: [39.7; 59.7] vs. 63.6%; 95% CI: [56.3; 70.0]; *p* < 0.0001). The 5-year OS, DMFS, and LRFS rates were 75.8% (95% CI: [70.1; 80.6]), 61.4% (95% CI: [55.4; 66.9]), and 71.1% (95% CI: [65.4; 76.0]), respectively. TN tumours were significantly associated with a poorer 5-year rate of OS (56.9%; 95% CI: [45.3; 66.9] vs. 84.0%; 95% CI: [77.6; 88.7]; *p* < 0.0001) compared to HR+ and/or HER2+ tumours. Furthermore, TN tumours, compared to HR+ and/or HER2+ tumours, were more at risk of distant metastases (DMFS at 53.9%; 95% CI: [43.3; 63.4] vs. 65.5%; 95% CI: [58.1; 71.8]; *p* = 0.0005) and locoregional recurrence (LRFS at 50.3%; 95% CI: [39.8; 60.0] vs. 80.5%; 95% CI: [74.1; 85.5]; *p* < 0.0001) (Figure 1).

### 3.4. Impact of Pathological Responses on DFS by Subtype

We evaluated the impact of pathological responses on tumours and lymph nodes with the Sataloff classification (Sataloff T and Sataloff N), according to HR and HER2 statuses. The impact of Sataloff T on DFS was significantly different according to HR/HER2 status (HER2+ and/or HR+ versus TN interaction test *p* = 0.0034). The interaction test was also significant for Sataloff N and HR/HER2 status (*p* = 0.025). Therefore, we reported all of the results and graphs on DFS according to these two subgroups: TN versus HR+ and/or HER2+ tumours. Appendix A illustrates DFS according to Sataloff T, Sataloff N, and HR/HER2 status.

### 3.5. Impact of Tumour and Node Responses on DFS According to Sataloff’s Classification

Table 3 presents the univariate analyses stratified by HR/HER2 status in accordance with the interaction test results. In both groups (HR+ and/or HER2+ vs. TN), we did not identify any difference in DFS between TA and TB responses. Additionally, we did not observe any difference in DFS between NA and NB responses. Therefore, we split the patients by TA-TB versus TC-TD responses and NA-NB versus NC-ND responses. Compared to Sataloff TA-TB, Sataloff TC-TD was associated with poorer DFS in the TN group and in the HR+ and/or HER2+ groups (Figure 2). Moreover, compared to Sataloff NA-NB, Sataloff NC-ND was associated with a lower DFS in the TN group (HR: 7.69 [95% CI: 3.53; 16.75]) but not in the HR+ and/or HER2+ group (HR: 1.64 [95% CI: 0.99; 2.69]).

After multivariate analyses (Table 4), only the Sataloff TC-TD response remained significantly associated with poorer DFS in the HR+ and/or HER2+ groups (HR: 1.85 [95% CI: 1.10; 3.11]), whereas NC-ND remained the only significant prognostic factor in the TN group (HR: 6.06, 95% CI: [2.59; 14.2]).

For HR+ and/or HER2+ patients, bootstrap validation confirmed that the sole significant negative prognostic factor for DFS was Sataloff TC-TD. No significant interaction was found between Sataloff T and N neither in HR+ and/or HER2+ patients (*p* = 0.29) nor in TN patients (*p* = 0.74). Within each Sataloff T group (TA-TB and TC-TD), Sataloff N was not significantly associated with DFS. For TN patients, bootstrap validation confirmed that Sataloff NC-ND (compared to NA-NB) was the sole factor significantly associated with DFS, with no significant impact of Sataloff T within each group of Sataloff N.

The association between overall survival and Sataloff response by breast cancer subtype is presented in the Appendix A. These results were consistent with those shown for DFS.

## 4. Discussion

In our large, retrospective, multicentre study, we showed that the 5-year DFS for patients with TN-IBC was poorer than the one for patients with HR+ and/or HER2+ IBC (45.8% vs. 65.2%; *p* < 0.001). Nevertheless, we showed that this poor prognosis was reversed by the lymph node response after neoadjuvant chemotherapy. The prognosis of patients with TN-IBC with the Sataloff NA-NB response appeared to be at least as good as patients with HR+ and/or HER2+ IBC.

The impact of the pathological complete response appeared to have a more substantial positive effect on DFS in TN-IBC than in HR+ and/or HER2+ IBC.

Van Uden et al. [15] recently published overall survival in 1061 patients with IBC diagnosed between 2006 and 2015. After a median follow-up of 2.4 years, the 5-year overall survival was 55.6% and differed significantly between subtypes, with the worst OS for TN-IBC (38.8%). Therefore, their overall and TN-specific 5-year OS rates were poorer than our study’s. Similarly, in 5265 patients with IBC diagnosed between 2012 and 2016, Grova et al. showed that, after a median follow-up at 26.8 months, there was a concordant 5-year rate overall survival at 51.6% above all of the subtypes and 34.4% for TN [9]. Thus, we investigated several hypotheses that may explain our better overall survival compared to other studies.

First, 309 (97.5%) patients in our study received trimodal treatment (chemotherapy, surgery, radiotherapy), whereas the corresponding rates in studies by Van Uden and Grova were 52.8% and 66%, respectively [9,15]. This larger proportion may explain our better survival outcomes. Indeed, Rueth et al. [21] recently demonstrated that 5- and 10-year survival rates were higher among those receiving trimodal treatment in a population of 10,197 patients with IBC who underwent surgical treatment between 1998 and 2010, compared with other strategies.

The more recent period of analysis was chosen to assess IBC outcomes in the modern era. The better outcomes possibly reflect the effects of treatment advances (diagnosis, types of systemic treatments, radiotherapy modality, surgical management, and multimodal treatments with supportive care, among other advances). However, there were at least as many TN-IBCs (31.3%) in our series than in other studies (Van Uden [24.3%] or Grova [26%] studies]).

The pathological complete response impact on OS and DFS has been previously demonstrated in IBC [13,15]. In patients who underwent neoadjuvant chemotherapy followed by surgical resection, we found pCR rates of 30.4% with Sataloff’s classification and 26.5% with the ypTNM system. Moreover, we observed a better rate of pCR with Sataloff than with ypTNM. The Sataloff definition of breast complete response (defined by TA) corresponds to total or near total response in the breast, which is not equivalent to ypT0/Tis. Cortazar et al. evaluated 11,955 patients with IBC and non-IBC breast cancer from 12 international trials between 1990 and 2011 and defined pCR using ypT0/is and ypN0. The authors found a 16% pCR rate in 482 patients with IBC [13]. With the same definition of pCR, Van Uden found a similar rate of pCR (16%) in 670 patients with an evaluated response [15]. Furthermore, Grova evaluated pCR in 3167 patients with IBC and found that pCR occurred in 20% of all women [9].

With the ypTNM system, our pCR rate (26.5%, defined as ypT0/Tis, N0) seems to be better than other studies. This could be explained by the extensive use of anthracyclines and taxanes, by the quality of supportive care which may have reduced the frequency of treatment interruptions or dose reductions, and by the selection of patients with neoadjuvant chemotherapy (rather than hormone therapy alone for example). This good pCR rate could explain our better survival results.

Previous studies have shown an impact of pCR on survival but have not identified different prognostic groups among patients without pCR. To our knowledge, this is the first study evaluating the prognostic value of tumour and node responses in a large cohort of TN-IBC. The use of Sataloff’s classification to assess outcomes in non-pCR is consistent with the BIG-NABCG recommendations [22]. For TN patients, multivariate analyses and bootstrap validation showed that Sataloff NC-ND (compared to NA-NB) was the only factor that was significantly associated with DFS (HR: 6.06) with no significant impact of Sataloff T within each group of Sataloff N.

The limitations of this study are those of any retrospective study. For example, pathologic response and scoring of different systems (ypTNM, Sataloff) after neoadjuvant chemotherapy was reported by individual institutions and was not reviewed by a central pathology team. The RCB index published by Symmans et al. in 2007 had not been incorporated into routine practice between 2010 and 2017. Due to substantial missing data, we could not describe its prognostic value in our population. Moreover, the median follow-up time for patients was 50 months, which may be considered a relatively short period for HR/HER+ IBC (but not for TN-IBC) and was longer than in other studies [9,15]. However, our overall survival results are limited by the small number of events, thus decreasing the statistical power to identify predictors. Nevertheless, our findings on overall survival seemed to be concordant with those on disease-free survival.

Finally, recently published studies have demonstrated the benefit of adjuvant treatments when pCR was not achieved. In a study conducted on Asian patients without pCR, Masuda et al. shown that adding 6–8 cycles of capecitabine resulted in improvements in DFS and OS, particularly in TN tumours [23]. For patients with HER2-positive early breast cancer who had a residual invasive disease, von Minckwitz et al. demonstrated that the risk of recurrence or death was 50% lower with adjuvant TDM-1 than with trastuzumab alone [24]. Despite the recent inclusion period (2010–2017), our study could not account for the impacts of these new treatment strategies.

Overall, our study has important clinical implications and could help further optimise trial designs. As mentioned above, patients are usually separated into two groups (pCR versus no pCR) to recommend adjuvant treatment. To our knowledge, there is no difference in adjuvant treatment strategy in patients with incomplete pathological response based on tumour and lymph node response independently.

Based on our results, we believe that patients with TN-IBC without pCR should be separated into two prognostic groups to optimise adjuvant treatment. On the one hand, patients with better prognoses (no residual lymph node disease and breast residual disease: Sataloff NA, NB, and TB, TC, TD), on the other hand, patients with poorer prognoses (residual lymph node disease, Sataloff NC, ND regardless of tumour response) who may benefit more from an intensification of adjuvant treatment.

Finally, considering the negative prognostic impact of residual lymph node disease in TN-IBC patients, we hypothesize that intensifying neoadjuvant treatment in all patients with a triple-negative IBC by improving the lymph node response rate could improve survival in this subtype.

For TN breast cancer patients, the recently published results of KEYNOTE-522 trial (addition of pembrolizumab to neoadjuvant chemotherapy) and BrighTNess trial (addition of carboplatin to neoadjuvant chemotherapy) are very encouraging. They will possibly lead to a new standard of care [25,26,27]. A specific trial evaluating the addition of immunotherapy to neoadjuvant chemotherapy in IBC is currently enrolling [28].

## 5. Conclusions

In conclusion, despite improvements in survival outcomes, IBC remains associated with a poor prognosis, especially for TN tumours. Identifying prognostic factors is the first step in proposing further prospective studies with new therapeutic strategies for selected patients. Currently, there is no difference in adjuvant treatment strategy in patients with incomplete pathological response based on tumour and lymph node responses independently. By using the Sataloff N classification, we identified a subgroup of TN-IBC patients without a pathological complete node response for whom adjuvant treatment intensification (such as treatment with concomitant and adjuvant immunotherapy + chemotherapy in the BREASTIMMUNE 03 Trial NCT03818685 or PARP inhibitors in the RADIOPARP phase 1 trial [29]) may be worth investigating. Furthermore, we identified a subgroup of TN-IBC patients without residual node disease with at least as good a prognosis as patients with HR+ and/or HER2+ IBC.

## Figures and Tables

**Figure 1 cancers-14-03928-f001:**
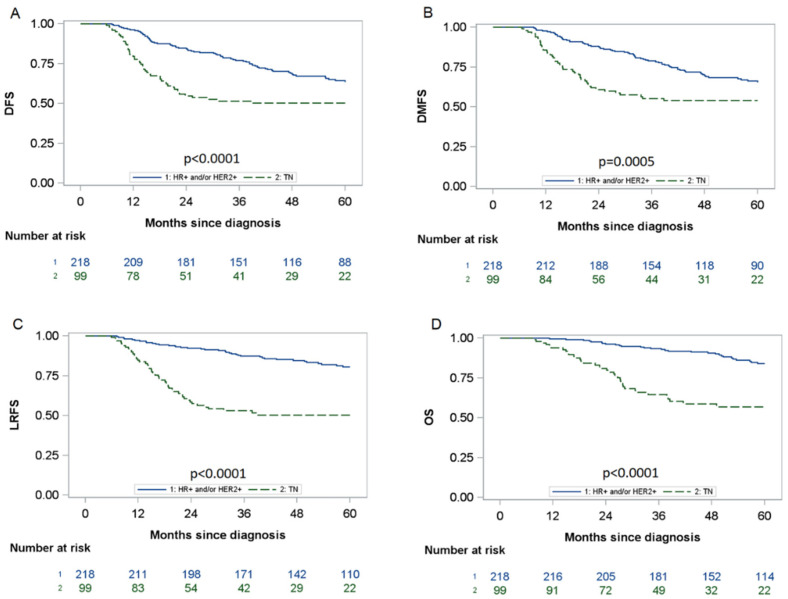
DFS, DMFS, LRFS and OS by triple-negative status. The blue line corresponds to patients with HR+ and/or HER2+ tumours, whereas the green line corresponds to patients with triple-negative (TN) tumours. (**A**) 5-year disease-free survival by breast cancer subtype; (**B**) 5-year distant metastasis-free survival by breast cancer subtype; (**C**) 5-year locoregional relapse-free survival by breast cancer subtype; (**D**) 5-year overall survival by breast cancer subtype.

**Figure 2 cancers-14-03928-f002:**
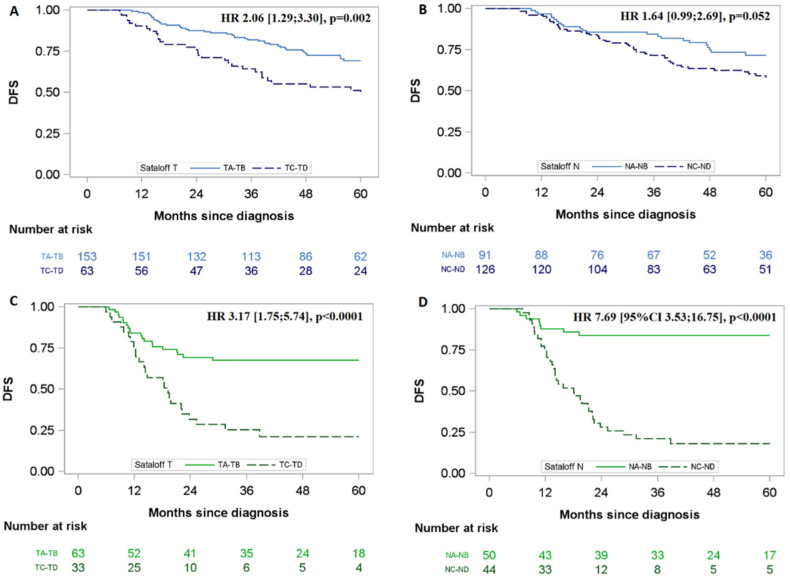
Association between DFS and Sataloff response by triple-negative status. The blue line corresponds to patients with HR+ and/or HER2+ tumours, whereas the green line corresponds to patients with triple-negative (TN) tumours. (**A**) Association between DFS and Sataloff TA-TB versus TC-TD response in patients with HR+ and/or HER2+ tumours: HR: 2.06 [95% CI: 1.29; 3.30], *p* = 0.002; (**B**) Association between DFS and Sataloff NA-NB versus NC-ND response in patients with HR+ and/or HER2+ tumours: HR: 1.64 [95% CI: 0.99; 2.69], *p* = 0.052; (**C**) Association between DFS and Sataloff TA-TB versus TC-TD response in patients with triple-negative tumours: HR: 3.17 [95% CI: 1.75; 5.74], *p* < 0.0001; (**D**) Association between DFS and Sataloff NA-NB versus NC-ND response in patients with triple-negative tumours: HR: 7.69 [95% CI 3.53; 16.75], *p* < 0.0001.

**Table 1 cancers-14-03928-t001:** Baseline demographics, clinical characteristics, and response after neoadjuvant chemotherapy.

	All N = 317	HR+ and/or HER2+ N = 218	HR- HER2- N = 99	*p*-Value
Age, years	53.0 (45.2–61.4)	53.0 (46.1–61.4)	51.8 (42–61.8)	0.37
Age, years (category)				0.52
<40	45 (14.3%)	27 (12.4%)	18 (18.4%)	
40–49	85 (27.0%)	58 (26.7%)	27 (27.6%)	
50–59	90 (28.6%)	65 (30.0%)	25 (25.5%)	
≥60	95 (30.2%)	67 (30.9%)	28 (28.6%)	
NA	2 (0.6%)	1 (0.5%)	1 (1.0%)	
Body Mass Index, kg/m^2^				0.040
<25	103 (32.6%)	61 (28.1%)	42 (42.4%)	
≥25 <30	105 (33.2%)	76 (35.0%)	29 (29.3%)	
≥30	108 (34.2%)	80 (36.9%)	28 (28.3%)	
NA	1 (0.3%)	1 (0.5%)	0	
Histology				0.15
Ductal	280 (88.9%)	188 (87%)	92 (92.9%)	
Lobular	18 (5.7%)	16 (7.4%)	2 (2.0%)	
Other	17 (5.4%)	12 (5.6%)	5 (5.1%)	
NA	2 (0.6%)	2 (0.9%)	0	
Grade SBR				<0.0001
1–2	137 (44.1%)	115 (53.5%)	22 (22.9%)	
3	174 (56.0%)	100 (46.5%)	74 (77.1%)	
NA	6 (1.9%)	3 (1.4%)	3 (3.0%)	
cN				0.047
N0	51 (16.2%)	41 (19.0%)	10 (10.1%)	
N+	264 (83.8%)	175 (81.0%)	89 (89.9%)	
NA	2 (0.6%)	2 (0.9%)	0	
WHO performance status				0.91
0	263 (83.5%)	180 (83%)	83 (84.7%)	
1	49 (15.6%)	35 (16.1%)	14 (14.3%)	
2–3	3 (1.0%)	2 (0.9%)	1 (1.0%)	
NA	2 (0.6%)	1 (0.5%)	1 (1.0%)	
Ki67				0.0002
<10%	3 (1.8%)	3 (2.5%)	0	
10–30%	62 (36.3%)	55 (45.5%)	7 (14.0%)	
>30%	106 (62.0%)	63 (52.1%)	43 (86.0%)	
NA	146 (46.1%)	97 (44.5%)	49 (49.5%)	
Sataloff T				0.064
TA	116 (37.2%)	74 (34.3%)	42 (43.8%)	
TB	100 (32.1%)	79 (36.6%)	21 (21.9%)	
TC	74/(23.7%)	50 (23.2%)	24 (25.0%)	
TD	22 (7.1%)	13 (6.0%)	9 (9.4%)	
Not available	5 (1.6%)	2 (0.9%)	3 (3.0%)	
Sataloff N				0.096
NA	97 (31.2%)	65 (30.0%)	32 (34.0%)	
NB	44 (14.2%)	26 (12.0%)	18 (19.2%)	
NC	116 (37.3%)	90 (41.5%)	26 (27.7%)	
ND	54 (17.4%)	36 (16.6%)	18 (19.2%)	
Not available	6 (1.9%)	1 (0.5%)	5 (5.1%)	
ypN				NC
N0	141 (44.5%)	90 (41.3%)	51 (51.5%)	
N1	71 (22.4%)	51 (23.4%)	20 (20.2%)	
N2	75 (23.7%)	56 (25.7%)	19 (19.2%)	
N3	25 (7.9%)	18 (8.3%)	7 (7.1%)	
Nx	5 (1.6%)	3 (1.4%)	2 (2.0%)	
ypT				NC
ypT0	72 (22.7%)	42 (19.3%)	30 (30.3%)	
ypTis	19 (6.0%)	13 (6.0%)	6 (6.1%)	
ypT1	78 (24.6%)	59 (27.1%)	19 (19.2%)	
ypT2	69 (21.8%)	47 (21.6%)	22 (22.2%)	
ypT3	42 (13.2%)	30 (13.8%)	12 (12.1%)	
ypT4	27 (8.5%)	19 (8.7%)	8 (8.1%)	
ypTx	10 (3.2%)	8 (3.7%)	2 (2.0%)	
Pathological complete response according to Sataloff	95 (30.4%)	60 (27.8%)	35 (36.5%)	0.12
Not available	5 (1.6%)	2 (0.9%)	3 (3.0%)	
Pathological complete response according to ypTNM	84 (26.5%)	51 (23.4%)	33 (33.3%)	0.063
Not available	0	0	0	

Data are median (IQR) or *n*/N (%). Percentages may not total 100 because of rounding. *p*-values were calculated using the χ^2^ test, Fisher’s exact test or Student *t* test. NA = Not available. HR = Hormonal Receptor. HER2 = Human Epidermoid growth factor Receptor-2. TN = Triple-Negative. NC = not computed. SBR = Scarff Blood Richardson. cN = clinical lymph Node. WHO = World Health Organization.

**Table 2 cancers-14-03928-t002:** Chemotherapy, surgery and radiotherapy details.

	All N = 317	HR+ and/or HER2+ N = 218	TN N = 99	*p*-Value
**Neoadjuvant chemotherapy protocol**	317 (100%)	218 (100%)	99 (100%)	NC
Number of cycles	8 (7–8)	8 (7–8)	8 (7–12)	0.014
(F)EC-T	233 (73.5%)	169 (77.5%)	64 (64.7%)	0.016
AC-T	34 (10.7%)	24 (11%)	10 (10.1%)	0.81
Taxanes received	316 (99.7%)	217 (99.5%)	99 (100%)	NC
Anthracyclines received	284 (89.6%)	193 (88.5%)	91 (91.9%)	0.36
Platinium salts received	3 (1%)	0	3 (3%)	NC
Trastuzumab alone	77 (24.3%)	75 (34.4%)	2 (2%)	<0.0001
Trastuzumab + Pertuzumab	14 (4.4%)	14 (6.4%)	0	
**Adjuvant systemic treatment**	219 (69.1%)	204 (93.6%)	15 (15.2%)	<0.0001
Hormonotherapy	168 (53%)	164 (75.2%)	4 (4%)	<0.0001
Trastuzumab	87 (27.4%)	85 (39%)	2 (2%)	<0.001
Adjuvant chemotherapy	10 (3.2%)	1 (0.5%)	9 (9.1%)	<0.0001
TDM-1	3 (1%)	3 (1.4%)	0	NC
Capecitabine	8 (2.5%)	1 (0.5%)	7 (7.1%)	0.0015
**Surgery**	317 (100%)	218 (100%)	99 (100%)	NC
Mastectomy + SLND	3 (0.9%)	1 (0.5%)	2 (2.0%)	NC
Mastectomy + ALND	310 (97.8%)	214 (98.2%)	96 (97.0%)	
Tumourectomy + SLND	0	0	0	
Tumourectomy + ALND	4 (1.3%)	3 (1.4%)	1 (1.0%)	
**Radiotherapy**	309 (97.5%)	215 (98.6%)	94 (95%)	0.11
Before surgery	18 (5.7%)	9 (4.1%)	9 (9.1%)	0.0028
After surgery	291 (91.8%)	206 (94.5%)	85 (85.9%)	
Dose (Gy)	50 (50–50)	50 (50–50)	50 (49–50)	0.0099
Fractions	25 (24–25)	25 (25–25)	25 (23–25)	0.039
Overall treatment time (days)	37 (37–40)	37 (35–41)	37 (35–40)	0.66
Target area				
B or CW alone ^a^	11 (3.5%)	7 (3.2%)	4 (4.1%)	0.74
B/CW + Level 2-3-4 ^a^	294 (93.9%)	206 (95.4%)	88 (90.7%)	0.11
Internal mammary node ^b^	211 (69.4%)	148 (69.8%)	63 (68.5%)	0.82
Level 1 ^b^	72 (23.7%)	52 (24.5%)	20 (21.7%)	0.60

Data are median (IQR) or *n*/N (%). Percentages may not total 100 because of rounding. *p*-values were calculated by using the χ^2^ test, Fisher’s exact test, Mann–Whitney U test, or Student T test. HR = Hormonal Receptor. HER2 = Human Epidermoid growth factor Receptor-2. TN = Triple-Negative. NC = No Computed. (F)EC-T = epirubicin with cyclophosphamide (+/− 5-Fluoro-uracil) plus docetaxel/paclitaxel. AC-T = doxorubicin with cyclophosphamide plus docetaxel/paclitaxel. TDM-1 = trastuzumab-emtansine. SLND = Sentinel Lymph Node Dissection. ALND = Axillary Lymph Node Dissection. B = Breast. CW = Chest Wall. Level 1–4 = regional lymph node areas. ^a^ 4 missing data. ^b^ 13 missing data.

**Table 3 cancers-14-03928-t003:** Univariate analyses of prognostic factors for disease-free survival (DFS) stratified by triple-negative status.

		HR+ and/or HER2+, N = 218		TN, N = 99	
		HR and 95% CI	*p*-Value	HR and 95% CI	*p*-Value
Age, years (category)	<40	1		1	
	40–49	0.76 [0.36; 1.60]	0.46	0.62 [0.27; 1.41]	0.26
	50–59	0.98 [0.48; 2.00]	0.95	0.65 [0.29; 1.48]	0.30
	≥60	0.66 [0.31; 1.39]	0.27	0.60 [0.27; 1.34]	0.21
BMI, kg/m^2^	<25	1		1	
	≥25 <30	1.43 [0.77; 2.66]	0.26	1.35 [0.69; 2.62]	0.38
	≥30	1.33 [0.72; 2.45]	0.37	0.97 [0.48; 1.97]	0.94
Histology	Ductal	1		1	
	Lobular	0.76 [0.31; 1.90]	0.56	1.33 [0.18; 9.68]	0.78
	Other	0.47 [0.12; 1.92]	0.29	1.96 [0.61; 6.32]	0.26
SBR	1–2	1		1	
	3	0.95 [0.60; 1.51]	0.82	0.85 [0.43; 1.68]	0.64
cN	N0	1		1	
	N+	0.97 [0.55; 1.71]	0.90	1.51 [0.54; 4.21]	0.43
WHO performance status	0	1		1	
	1–3	1.10 [0.60; 2.01]	0.75	1.74 [0.86; 3.50]	0.12
Preoperative radiotherapy	No	1		1	
	Yes	1.84 [0.67; 5.05]	0.24	3.46 [1.66; 7.21]	0.0009
Sataloff T	TA	1		1	
	TB	1.12 [0.61; 2.05]	0.73	2.06 [0.86; 4.94]	0.11
	TC	2.05 [1.12; 3.77]	0.020	5.66 [2.64; 12.16]	<0.0001
	TD	2.79 [1.17; 6.64]	0.020	1.92 [0.60; 6.14]	0.27
Sataloff T	TA-TB	1		1	
	TC-TD	2.06 [1.29; 3.30]	0.002	3.17 [1.75; 5.74]	<0.0001
Sataloff N	NA	1		1	
	NB	1.02 [0.42; 2.49]	0.96	1.10 [0.26; 4.58]	0.90
	NC	1.44 [0.79; 2.62]	0.24	7.10 [2.65; 19.01]	<0.0001
	ND	2.29 [1.15; 4.53]	0.018	9.59 [3.44; 26.70]	<0.0001
Sataloff N	NA-NB	1		1	
	NC-ND	1.64 [0.99; 2.69]	0.052	7.69 [3.53; 16.75]	<0.0001

Cox proportional hazards regression analysis of progression-free survival. HR = hazard ratio. CI = confidence interval. HR = Hormonal Receptor. HER2 = Human Epidermoid growth factor Receptor-2. TN = Triple-Negative. NC = not computed. SBR = Scarff Blood Richardson. cN = clinical lymph Node. WHO = World Health Organization. Patients with TN tumours with a Sataloff NA-NB response (*n* = 50, 53%) did not have a worse disease-free survival than other patients with HR+ and/or HER2+ tumours (HR: 0.51 95% CI: [0.25; 1.06], *p* = 0.071), with 5-year DFS rates of 83.8% [95% CI: 70.1; 91.5] and 63.6% [95% CI: 56.3; 70.1], respectively.

**Table 4 cancers-14-03928-t004:** Multivariate analyses of prognostic factors for disease-free survival (DFS) stratified by triple-negative status.

HR+ and/or HER2+, N = 218		HR and 95% CI	*p*-Value
Sataloff T	TA-TB	1	
	TC-TD	1.85 [1.10; 3.11]	0.020
Sataloff N	NA-NB	1	
	NC-ND	1.3 [0.75; 2.24]	0.35
**TN, N = 99**		HR and 95% CI	*p*-value
Sataloff T	TA-TB	1	
	TC-TD	1.33 [0.69; 2.54]	0.39
Sataloff N	NA-NB	1	
	NC-ND	6.06 [2.59; 14.2]	<0.0001
Preoperative radiotherapy	No	1	
	Yes	1.82 [0.85; 3.89]	0.12

Factors with *p*-value less than 0.1 in univariate analyses were included in multivariate Cox proportional hazards regression analysis of progression-free survival. HR = hazard ratio. CI = confidence interval. HR = Hormonal Receptor. HER2 = Human Epidermoid growth factor Receptor-2. TN = Triple-Negative.

## Data Availability

All data generated and analysed during this study are included in this published article (and its Appendix A). The data are available upon request from the corresponding author. The data are not publicly available for confidentiality reasons.

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
