# Peer review of "Different Prognostic Values of Tumour and Nodal Response to Neoadjuvant Chemotherapy Depending on Subtypes of Inflammatory Breast Cancer, a 317 Patient-Study"

_cancers, 2022, doi:10.3390/cancers14163928_

Round 1
Reviewer 1 Report
Interesting article. I think you should only emphasize the role of neoadjuvant therapy in more advanced breast cancers regardless their molecular status
Author Response
Thanks for the comment.
Emphasising the role of neoadjuvant therapy in more advanced breast cancers regardless of their molecular status could have been an option. However, we preferred to focus on the triple-negative subpopulation. Inflammatory breast cancer has a poor prognosis, especially in its triple-negative form. For these patients, treatment options are limited.
Recently, many studies have been conducted on the triple-negative subtype with good results. Nevertheless, there is currently limited specific data on inflammatory breast cancer. We hope our study can contribute to the research dynamism for these young patients.
Reviewer 2 Report
The manuscript presents a large, multicentre, retrospective study that analyzes the survival outcomes of patients with nonmetastatic inflammatory breast cancer. A total of 317 patients were included, which makes this study respectable considering the rarity of this type of cancer. The authors aimed to evaluate pathological response after neoadjuvant chemotherapy assessed using Sataloff's classification. They identified patients with residual node disease as a group with a poor prognosis with special meaning for triple-negative BC.
In summary, this is a thoroughly written paper based on a large consecutive nonselected cohort of patients, and the findings could be important for clinical practice. Several comments and questions below could help the authors improve the study's presentation.
1. How was the finding in the breast evaluated as inflammatory carcinoma? Based on histological examination only? Based on the clinical findings - breast inflammation? Inflammatory carcinoma is more of a clinical entity, the histological correlate of which may be the presence of tumor lymphangiopathy in the skin and subcutaneous tissue.
2. Were the patients evaluated by pathologists using both classifications (ypTNM and Sataloff), or was it retrospectively reclassified from the pathology report?
3. Ki-67 proliferation index was not assessed in many patients from core biopsies. Why?
4. A double-check of the manuscript is appropriate - language mistakes, typos (Ki76, sels,…), explanation of abbreviations,
5. Do you have any explanation why only three patients were treated with platinum salts in a neoadjuvant setting?
6. What cut-off for ER and PF was used for HR positivity/negativity? Please add to Methods.
7. What methodology was used for follow-up evaluation? Please add to Methods.
8. There is no definition of considered events in survival endpoints. Were the survival times calculated from the diagnosis of surgery? Were the deaths considered from all causes?
9. Supplementary material should be referred according to journal instructions.
10. All Kaplan-Meier curves are cut off in 60 months. It would be more appropriate to present the complete curves. Moreover, the titles of the figures are misleading (5-year survival outcomes).
11. There are no DFS events after approx. 40 months in TN-IBC subtypes. It is surprising in this aggressive form of BC.
12. The claim “Breast response was not identified as a significant prognostic factor in TN-IBC patients.” is too bold. What was the association between Sataloff T and N. Was considered the interaction between T and N response in multivariable analysis?
13. The surrogate of pCR for survival is known but is not immediately obvious from the results.
14. The finding of the negative prognostic impact of residual lymph node disease in TN-IBC patients is interesting but not directly applicable to the eventual intensification of neoadjuvant treatment. Check the discussion.
15. The authors describe the subgroup of patients with incomplete pathological response in the discussion, but this subgroup is not considered in the results section.
16. The title needs some rework, not to be specific to nodal response only.
Author Response
- How was the finding in the breast evaluated as inflammatory carcinoma? Based on histological examination only? Based on the clinical findings - breast inflammation? Inflammatory carcinoma is more of a clinical entity, the histological correlate of which may be the presence of tumor lymphangiopathy in the skin and subcutaneous tissue.
You are correct that IBC is a unique entity, and its diagnosis remains a challenge that has been the subject of many articles. Diagnosis of inflammatory breast cancer was usually based on clinical findings (rapid onset of symptoms, erythema, peau d’orange, breast oedema or swelling …). Breast cancer was always confirmed with a biopsy. Pathological confirmation of dermal lymphatic emboli (skin punch biopsies) was usually not performed, given that it is not required to confirm the diagnosis. We added this to the methods.
Jagsi, R., Mason, G., Overmoyer, B.A. et al. Inflammatory breast cancer defined: proposed common diagnostic criteria to guide treatment and research. Breast Cancer Res Treat 192, 235–243 (2022)
Robert H. Hester, Gabriel N. Hortobagyi, Bora Lim, Inflammatory breast cancer: early recognition and diagnosis is critical, American Journal of Obstetrics and Gynecology, Volume 225, Issue 4, 2021.
- Were the patients evaluated by pathologists using both classifications (ypTNM and Sataloff), or was it retrospectively reclassified from the pathology report?
The pathologists have used the Sataloff system in their reports. If the ypTNM score was not reported directly in the report, it was reclassified from the report’s details by the investigators.
- Ki-67 proliferation index was not assessed in many patients from core biopsies. Why?
Between 2010 and 2017, in France, there was no consensus to routinely perform the KI67 score since there was no therapeutic impact based on its results. It was very centre dependent. Nevertheless, we strongly agree that the KI67 score provided valuable insights into tumour aggressiveness.
- A double-check of the manuscript is appropriate - language mistakes, typos (Ki76, sels,…), explanation of abbreviations,
A company has proofread the article. We apologize for any remained typos or language mistakes. We have corrected several points:
- Explanation of abbreviation: UICC, AJCC, IRB, GDPR.
- KI67, Platinium salts
- Language mistakes
- Do you have any explanation why only three patients were treated with platinum salts in a neoadjuvant setting?
In France, between 2010 and 2017, platinum salts in a neoadjuvant setting for TN breast cancer were an option but were not the standard of care. The standard of care was four (F)EC and four cycles of taxanes.
Early Breast Cancer: ESMO Clinical Practice Guidelines for diagnosis, treatment and follow-up. Ann Oncol. 2019;30(8):1194-1220. F. Cardoso, S. Kyriakides, S. Ohno, et. al, on behalf of the ESMO Guidelines Committee
- What cut-off for ER and PF was used for HR positivity/negativity? Please add to Methods.
Thank you, this is a critical cut-off that we forgot to precise.
Local pathologists defined the hormone receptor status with immunohistochemistry (deemed positive if oestrogen receptor, progesterone receptor, or both were ≥10%).
It is added to the methods.
- What methodology was used for follow-up evaluation? Please add to Methods.
Regular follow-up visits were performed every six months from the end of radiotherapy until at least five years. A Clinical examination and annual contralateral mammography +/- ultrasound were performed. Whole body imaging tests were not systematically done in asymptomatic patients.
It is added to the methods.
- There is no definition of considered events in survival endpoints. Were the survival times calculated from the diagnosis of surgery? Were the deaths considered from all causes?
Locoregional recurrence was defined as the recurrence of disease in the ipsilateral chest wall, skin, muscle, or in the ipsilateral axillary, supraclavicular, infraclavicular, or internal mammary lymph nodes. All other recurrences were classified as distant metastases, including the spread of the disease to the contralateral breast.
Disease-free survival was defined as the time from diagnosis (date of the first biopsy) to recurrence (locoregional and/or distant metastases) or death (all causes).
I add these precisions to the manuscript.
- Supplementary material should be referred according to journal instructions.
We apologize for this. Thank you for the comment; changes have been made in compliance with your comment and journal instructions.
- All Kaplan-Meier curves are cut off in 60 months. It would be more appropriate to present the complete curves. Moreover, the titles of the figures are misleading (5-year survival outcomes).
The median follow-up time was 52.5 months. Consequently, the number of patients followed after 5 years is low, especially in the TN group (n=22), which would lead to a poor precision of estimates.
Thank you, I made the modifications.
- There are no DFS events after approx. 40 months in TN-IBC subtypes. It is surprising in this aggressive form of BC.
Thank you, this is a good observation.
The aggressiveness of the triple-negative subtype can be clearly seen in Figure 1. We can see that a large proportion of patients relapsed (especially in the metastatic form) during the first 3 years (steep curve for the TN subtype). After that, it seems that patients relapsed less (more horizontal curves). These results are consistent with several publications presenting a similar aspect of the curves for the TN subtype.
Furthermore, patients who met our inclusion criteria and could receive neoadjuvant chemotherapy probably had a good prognosis.
van Uden DJP, van Maaren MC, Bult P, Strobbe LJA, van der Hoeven JJM, Blanken-Peeters CFJM, et al. Pathologic complete response and overall survival in breast cancer subtypes in stage III inflammatory breast cancer. Breast Cancer Res Treat. 2019 Jul;176(1):217–26.
Biswas T, Jindal C, Fitzgerald TL, Efird JT. Pathologic Complete Response (pCR) and Survival of Women with Inflammatory Breast Cancer (IBC): An Analysis Based on Biologic Subtypes and Demographic Characteristics. Int J Environ Res Public Health. 2019 Jan 4;16(1):124.
- The claim “Breast response was not identified as a significant prognostic factor in TN-IBC patients.” is too bold. What was the association between Sataloff T and N. Was considered the interaction between T and N response in multivariable analysis ?
We fully agree that “Breast response was not identified as a significant prognostic factor in TN-IBC patients” is too bold.
And we proposed the following modification: “Breast response does not appear to be a decisive prognostic factor in patients with TN-IBC compared to lymph node response.”
The association between Sataloff T and N was statistically significant in HR+ and/or HER2+ patients (p<0.001) and in TN patients (p<0.001). No significant interaction was found between Sataloff T and N neither in HR+ and/or HER2+ patients (p=0.29) nor in TN patients (p=0.74).
This result is in line with the results of the exploratory analysis mentioned at the end of the result section:
“Within each Sataloff T group (TA-TB and TC-TD), Sataloff N was not significantly associated with DFS. For TN patients, bootstrap validation confirmed that Sataloff NC-ND (compared to NA-NB) was the sole factor that was significantly associated with DFS, with no significant impact of Sataloff T within each group of Sataloff N.”
I add these precisions at the end of the results.
Nevertheless, we can not exclude that these results could not be modified in a larger population since the interaction analyses required many patients to be powerful. Consequently, it is a good thing to adjust our claim.
- The surrogate of pCR for survival is known but is not immediately obvious from the results.
You are right,
Several publications have already demonstrated the impact of pCR on survival outcomes in IBC patients. In our study, we chose to ask the question differently, investigating the survival impact of breast response on the one hand and the other hand of lymph node response.
van Uden, D.J.P.; van Maaren, M.C.; Bult, P.; Strobbe, L.J.A.; van der Hoeven, J.J.M.; Blanken-Peeters, C.F.J.M.; Siesling, S.; de Wilt, J.H.W. Pathologic Complete Response and Overall Survival in Breast Cancer Subtypes in Stage III Inflammatory Breast Cancer. Breast Cancer Res Treat 2019, 176, 217–226, doi:10.1007/s10549-019-05219-7.
- The finding of the negative prognostic impact of residual lymph node disease in TN-IBC patients is interesting but not directly applicable to the eventual intensification of neoadjuvant treatment. Check the discussion.
This is the part of the text to which you refer:
“Finally, considering the negative prognostic impact of residual lymph node disease in TN-IBC patients, it appears necessary to also intensify neoadjuvant treatment in order to achieve the best possible pCR rates, which might translate into improved survival outcomes in these patients.”
You are right, the survival benefit of intensification of neoadjuvant treatment in TN-IBC is a hypothesis based on our results. We have demonstrated the negative impact of residual lymph node disease (Sataloff NC-ND) after neoadjuvant chemotherapy in TN-IBC. We hypothesize that intensifying neoadjuvant treatment by improving the lymph node response rate could improve survival in this subpopulation.
We proposed the following modifications:
“Finally, considering the negative prognostic impact of residual lymph node disease in TN-IBC patients, we hypothesize that intensifying neoadjuvant treatment by improving the lymph node response rate could improve survival in this subpopulation.”
- The authors describe the subgroup of patients with incomplete pathological response in the discussion, but this subgroup is not considered in the results section.
This is the part of the text to which you refer:
“Overall, our study has important clinical implications and could help in the further optimization of trial design. As mentioned above, patients are usually separated into two groups (pCR versus no pCR) to recommend adjuvant treatment. To our knowledge, there is no distinction that is made in the treatment strategy for patients with incomplete pathological response regarding tumour and node responses. Patients with TN-IBC without pCR should be separated into two different prognostic groups after neoadjuvant chemotherapy: patients with poorer prognoses (absence of complete nodal response: NC, ND according to Sataloff classification, regardless of tumour response) who may benefit more from an intensification of adjuvant treatment, as opposed to patients with a complete node response and breast residual disease (NA, NB, and TB, TC, TD)”
You are right. We did not discuss patients with incomplete pathological response in the results section. Here we try to show the clinical relevance of our results and explore which therapeutic changes could be made considering the current practice.
Currently, there is no difference in adjuvant treatment strategy in patients with incomplete pathological response based on tumour and lymph node response independently. In TN subtype with residual disease after neoadjuvant chemotherapy, we can propose capecitabine in adjuvant setting. For example, there is no distinction made between one patient with a complete breast response and residual lymph node disease and a patient with a residual breast disease without lymph node disease.
Based on our results, we believe that TN-IBC patients with residual lymph node disease would require intensification of adjuvant treatment (immunotherapy? PARP inhibitors?)
We proposed the following modifications:
“Overall, our study has important clinical implications and could help further optimise trial designs. In the current clinical practice, patients are usually separated into two groups (pCR versus no pCR) to recommend adjuvant treatment. To our knowledge, there is no difference in adjuvant treatment strategy in patients with incomplete pathological response based on tumour and lymph node response independently.
Based on our results, we believe that patient with TN-IBC without pCR should be separated into two prognostic groups to optimise adjuvant treatment. On the one hand, patients with better prognoses (no residual lymph node disease and breast residual disease: Sataloff NA, NB, and TB, TC, TD), on the other hand, patients with poorer prognoses (residual lymph node disease, Sataloff NC, ND regardless of tumour response) who may benefit more from an intensification of adjuvant treatment.”
- The title needs some rework, not to be specific to nodal response only.
Good suggestion,
It could be: “Different prognostic values of tumour and nodal responses to neoadjuvant chemotherapy depending on subtypes of inflammatory breast cancer, a 317 patient-study”
Instead of
“Different prognostic values of nodal response to neoadjuvant chemotherapy depending on subtypes of inflammatory breast cancer, a 317 patient-study”
Reviewer 3 Report
Referee report for Cancers - ”Different prognostic values of nodal response to neoadjuvant chemotherapy depending on subtypes of inflammatory breast cancer, a 317 patient-study”
Neoadjuvant therapies are able to achieve a considerable optimization of patients’ healthcare. In particular the criteria that have to be imposed in a homogeneous manner in different institutions are currently targeted by a widespred research.
The paper is based on a multicentric study, which represents the important way to obtain a rich dataset. Moreover a particular attention is focused on the requirement for a central pathologic team that should impose the absence of individual institutions criteria. The results concern the primary role that has to be assigned to nodal response for triple negative inflammatory breast cancer. A well posed comparison with studies referred to different therapeutic plans adopted in the past underlines the efficacy of the new therapies. The paper might be suitable for publication in Cancers. However, some clarifications are necessary and some issues must be addressed.
1. in subsection 2.3 Statistical analysis you report the use of recursive partitioning analysis (RPA) for exploratory analysis, which is known to be afftected by overfitting: why did you miss the implementation of a more stable algorithm as random forest? Moreover you should specify the software;
2. in subsection 3.3. Follow-up you report two values for the median, that are not referred explicitly to any patients’ group;
3. in Table 1 you should add the number of NA (not a number) in each feature, such that you can delete the division and report just the number of patients and the percentage (e.g. Age, < 40 : 45(14.3%));
4. in Table 2 you have to act in the same way, eventually adding a footnote for low number of cases where the total number is different; 5. typos (please take care in future of the font): lines 23, 33-34, 43, 44, 47, 56, 57 (parenthesis), 68, 73, 83, 88-89, 90, 108, 116 (delete one ”Wilk”), 116-117, 118, 119, 120-121, 123 (should it be ”group label”?), 131, 132, 142, 151, 224, 230, 233, 234, 236, 243, 245, 247, 257, 266, 303, 306-308, 310-311, 312-314, 342, 343-344.
Author Response
In subsection 2.3 Statistical analysis you report the use of recursive partitioning analysis (RPA) for exploratory analysis, which is known to be afftected by overfitting: why did you miss the implementation of a more stable algorithm as random forest? Moreover you should specify the software;
We fully agree that random forest is a most stable algorithm in the case of exploratory analyses with a high number of parameters. In our study, we used RPA analysis to investigate if Sataloff N could play a role within each Sataloff T group (TA-TB and TC-TD) for HR+ and/or HER2+ patients. The same analysis was performed for TN patients to investigate the impact of Sataloff T and preoperative radiotherapy within each group of Sataloff N. In others words, only the significant parameters in the multivariate analyses were analysed, thus limiting the number of parameters. And as already mentioned in the method section, to ensure the stability of the results, a bootstrap validation was performed. The method section was slightly modified to clarify which parameters were involved.
Software used for statistical analyses is mentioned in the article “All of the statistical analyses were performed by using SAS software v9.4 (Institute Inc., Cary, NC 25513).”
In subsection 3.3. Follow-up you report two values for the median, that are not referred explicitly to any patients’ group;
We reported two values for median follow-up, one for the overall population and one for surviving patients (excluding deceased patients). Indeed, the median follow-up for all patients may appear low because of early deaths in this subpopulation of inflammatory breast cancer.
In Table 1 you should add the number of NA (not a number) in each feature, such that you can delete the division and report just the number of patients and the percentage (e.g. Age, < 40 : 45(14.3%))
Thank you for your comment. Changes have been made and make the tables more readily understandable.
In Table 2 you have to act in the same way, eventually adding a footnote for low number of cases where the total number is different;
Thanks for your advice. The Table 2 presentation is now more intelligible.
Typos (please take care in future of the font): lines 23, 33-34, 43, 44, 47, 56, 57 (parenthesis), 68, 73, 83, 88-89, 90, 108, 116 (delete one ”Wilk”), 116-117, 118, 119, 120-121, 123 (should it be ”group label”?), 131, 132, 142, 151, 224, 230, 233, 234, 236, 243, 245, 247, 257, 266, 303, 306-308, 310-311, 312-314, 342, 343-344.
Thanks for your feedback. I have corrected the font problems. The transformation by the editorial team of my manuscript written in “Calibri” to “Palatino Linotype” had some problems. Unfortunately, the line numbering system that you used does not correspond to the one in the article provided by the editor for the review. I apologise if I miss other typos. We will be particularly vigilant in the proof before publication if you judge that this work can be published.
Round 2
Reviewer 2 Report
In the reviewed manuscript, the authors have carefully considered all reviewers' recommendations for improving the paper. Some issues still remain and could be handled.
· Methods: Please add the statistical methodology of the follow-up estimation.
· Discussion: "…intensifying neoadjuvant treatment by improving the lymph node response rate could improve survival in this subpopulation". This is a good idea, but the intensifying would have to be done in all patients because the subgroup of patients (or some predictors thereof) with residual lymph node disease is unknown at the time of diagnosis. Modify or remove this part.
· Consider whether it would not be appropriate to add to the methods that Sataloff's classification was the standard evaluation used by pathologists in France during the study period.
Author Response
1) Methods: Please add the statistical methodology of the follow-up estimation.
Thank you for your comment. It's added to the methods section.
2) Discussion: "…intensifying neoadjuvant treatment by improving the lymph node response rate could improve survival in this subpopulation". This is a good idea, but the intensifying would have to be done in all patients because the subgroup of patients (or some predictors thereof) with residual lymph node disease is unknown at the time of diagnosis. Modify or remove this part.
You are right.
Sorry I didn't understand what you meant in the first round. I modified this part as follows :
"Finally, considering the negative prognostic impact of residual lymph node disease in TN-IBC patients; We hypothesize that intensifying neoadjuvant treatment in all patients with a triple-negative IBC, by improving the lymph node response rate could improve survival in this subtype."
3) Consider whether it would not be appropriate to add to the methods that Sataloff's classification was the standard evaluation used by pathologists in France during the study period.
I added this part in the methods section (procedures):
"We recorded the responses after neoadjuvant chemotherapy with two pathological response assessment systems (absolute and relative grading systems), as reported by pathologists at each participating centre (ypTNM [17] and Sataloff [18]). Sataloff's classification was the standard evaluation used by pathologists in France during the study period. Details of this classification are available in the supplementary materials (Table S1)."